# Healthcare Worker Study Cohort to Determine the Level and Durability of Cellular and Humoral Immune Responses after Two Doses of SARS-CoV-2 Vaccination

**DOI:** 10.3390/vaccines10111784

**Published:** 2022-10-24

**Authors:** Chiara Dentone, Daniela Fenoglio, Marta Ponzano, Matteo Cerchiaro, Tiziana Altosole, Diego Franciotta, Federica Portunato, Malgorzata Mikulska, Lucia Taramasso, Laura Magnasco, Chiara Uras, Federica Magne, Francesca Ferrera, Graziana Scavone, Alessio Signori, Antonio Vena, Valeria Visconti, Gilberto Filaci, Alessandro Sette, Alba Grifoni, Antonio Di Biagio, Matteo Bassetti

**Affiliations:** 1Infectious Disease Clinic, IRCCS Policlinico San Martino Hospital, Largo Rosanna Benzi, 10, 16132 Genoa, Italy; 2Department of Internal Medicine, Centre of Excellence for Biomedical Research (CEBR), University of Genoa, 16126 Genoa, Italy; 3Biotherapy Unit, IRCCS Policlinico San Martino, 16132 Genoa, Italy; 4Biostatistics Unit, Department of Health Science, University of Genova, 16132 Genova, Italy; 5Infectious Diseases Unit, Department of Health Science (DISSAL), University of Genoa, 16126 Genoa, Italy; 6Autoimmunology Laboratory, IRCCS Policlinico San Martino, 16132 Genoa, Italy; 7Laboratory Unit, IRCCS Policlinico San Martino, 16132 Genoa, Italy; 8Center for Infectious Disease and Vaccine Research, La Jolla Institute for Immunology (LJI), La Jolla, CA 92037, USA; 9Department of Medicine, Division of Infectious Diseases and Global Public Health, University of California, San Diego (UCSD), La Jolla, CA 92037, USA

**Keywords:** SARS-CoV-2, cellular and humoral immune responses, second dose, variants

## Abstract

We prospectively studied immunological response against SARS-CoV-2 after vaccination among healthcare workers without (group A) and with previous infection (group B). The analyses were collected at T0 (before the BNT162b2), T1 (before the second dose), T2 and T6 (1 and 6 months after the second dose). For cellular immune response, the activation-induced cell marker assay was performed with CD4 and CD8 Spike peptide megapools expressed as Stimulation Index. For humoral immune response, we determined antibodies to Spike-1 and nucleocapsid protein. The linear mixed model compared specific times to T0. The CD4+ Spike response overall rate of change was significant at T1 (*p* = 0.038) and at T2 (*p* < 0.001), while decreasing at T6. For CD8+ Spike reactivity, the interaction between the time and group was significant (*p* = 0.0265), and the *p* value for group comparison was significant at the baseline (*p* = 0.0030) with higher SI in previously infected subjects. Overall, the anti-S Abs significantly increased from T1 to T6 compared to T0. The group B at T6 retained high anti-S titer (*p* < 0.001). At T6, in both groups we found a persistent humoral response and a high CD4+ T cell response able to cross recognize SARS-COV-2 variants including epsilon, even if not a circulating virus at that time.

## 1. Introduction

During the implementation of the severe acute respiratory syndrome coronavirus 2 (SARS-CoV-2) vaccination strategy it was vital to understand how vaccination influences immune responses and protection among those who have had prior natural SARS-CoV-2 infection [1].

Immune memory against SARS-CoV-2 infection is associated with cellular and humoral adaptive immunity [2,3]. The generation and clinical importance of T cell responses following SARS-CoV-2 vaccination is also discussed, with particular focus on protection against viral variants. Evidence thus far indicates that T cells play a critical role in protection against SARS-CoV-2 [4]. Interestingly, Oberhardt et al. demonstrated that a protective clinical effect is seen within 11 days after the first vaccination, and a robust CD8+ T cell response can be seen in this early period [5]. T cell responses will also be needed to support the generation and maintenance of high-affinity antibodies, and dual vaccination with BNT162b2 leads to reliable induction of virus-specific CD4+ T cell responses [6]. As the prevalence of vaccination and natural infection increases across the world, there is increasing interest in developing approaches that predict individual risk of primary infection or reinfection. 

The duration and effectiveness of immunity from infection with and vaccination against SARS-CoV-2 are relevant to health policy interventions, including the timing of vaccine boosters [7], especially for healthcare workers (HCW) [8,9]. HCW were among the first eligible for vaccination, along with individuals at risk of coronavirus disease-19 (COVID-19). The association of vaccination with asymptomatic infection and transmission remains unclear, with important implications for public health policy. Data on vaccine effectiveness for HCW, who are at risk of exposure to SARS-CoV-2, are also limited [10,11]. Several studies have been conducted in Italy to assess behavioral changes during lockdowns and throughout the ongoing COVID-19 pandemic in the general population and in specific populations such as HCW [12]. Knowing whether and to what extent vaccine effectiveness wanes is crucial to establish the timing of booster doses and define target populations that should be prioritized for booster doses [13]. Finally, knowing the protection given to the vaccine is an important message for HCW managing SARS-CoV-2-infected patients. Several systematic reviews of SARS-CoV-2 vaccination efficacy and effectiveness studies have been published, but none have evaluated the duration of protection of COVID-19 [14,15,16,17,18].

In this prospective cohort analysis, we aimed to determine the level and durability of the cellular and humoral immune responses against SARS-CoV-2 infection after two doses of the vaccine in HCW cohort at our Infectious Disease Unit in University Hospital in Genoa, Italy (HCWAX study).

## 2. Materials and Methods

We prospectively enrolled HCW without (group A) and with (group B) previous SARS-CoV-2 infections working in our Infectious Diseases Unit, where we managed COVID-19 patients. We have consecutively enrolled healthcare professionals from our Infectious Disease Unit and 29 professionals, out of the 59 eligible, accepted to participate to the study (participation rate = 29/59 = 49%). Our reference population was the healthcare professionals of working age from our Infectious Disease Unit (Medical Doctors N = 21, Residents N = 12, Nurses, N = 24 and Physioterapists, N = 2).

We collected peripheral blood at baseline (T0, before the BNT162b2 vaccine), T1 (before the second dose), T2 and T6 (after 1 and 6 months after of the second dose, respectively). A previous infection was defined as a previous real-time reverse-transcriptase polymerase chain reaction (RT-PCR) positive result from a nasal and/or throat swab according to World Health Organization interim guidance [19]. Nasopharyngeal swabs were not performed routinely for screening in our cohort, but only in symptomatic subjects. The HCWAX study was carried out in accordance with the principles of the Declaration of Helsinki and approved by the National and Regional Ethic Committee (IRCCS Lazzaro Spallanzani Institut and Comitato Etico Regione Liguria, respectively) (CER Liguria 2712PRNO010221, 78/2021 11252– Approval 1320, 2021). Informed consent was signed by all the participants in the study.

To determine the level and durability of the cellular immune responses against SARS-CoV-2, the activation-induced cell marker assay (AIM) was performed using peptide megapools (MPs) containing SARS-CoV-2 Spike epitopes (MPs containing CD4 and CD8 Spike epitopes). 

### 2.1. Peptide Megapool Preparation

To identify SARS-CoV-2-specific T cell epitopes, 15-mer peptides overlapping by 10 amino acids and spanning entire SARS-CoV-2 proteins were synthesized. All peptides were synthesized as crude material (TC Peptide Lab, San Diego, CA, USA) and individually resuspended in dimethyl sulfoxide (DMSO) and pooled by the antigen of provenance into MPs followed by sequentially lyophilization and resuspension in DMSO at 1 mg/mL, as previously reported [20]. In particular, we used Spike peptide megapool (MPS) to evaluate the entire CD4+ and CD8+ T cell reactivity, as previously described [20].

### 2.2. PBMC Isolation 

Whole blood was collected from all donors in heparin-coated tubes. Peripheral blood mononuclear cells (PBMC) were isolated by density-gradient sedimentation using Ficoll-Paque (Euroclone). 

The plasma was then carefully removed from the cell pellet and stored at −20 °C to evaluate the level of SARS-CoV-2-specific antibodies. Isolated PBMC were resuspended in RPMI (Euroclone) supplemented with 10% heat-inactivated fetal calf serum (FCS, Euroclone) until being used in the assays.

### 2.3. Activation-Induced Cell Marker Assay

To perform activation-induced cell marker assay (AIM), 1 × 10^6^ PBMCs per well were cultured for 24 h in the presence of SARS-CoV-2-specific MPs [1 μg/mL], in 96-wells U bottom plates. As a negative control, an equimolar amount of Dimethyl Sulfoxide (DMSO) was used as a negative control, and phytohemagglutinin (PHA, Roche, 1μg/mL) was included as the positive control [21,22,23]. At the end of the incubation, the samples were stained for 20 min at the room temperature with the following fluorochrome-conjugated monoclonal antibodies (mAbs): anti-human CD3 Allophycocianin (APC)-R700 (UCHT1 clone, BD Biosciences San Diego, CA, USA), anti-human CD4 Brilliant Violet (BV) 605 (RPA-T4 clone, BD Biosciences), anti-human CD8 BV650 (RPA-T8 clone, BD Biosciences), anti-human CD14 V500 (M5E2 clone, BD Biosciences), anti-human CD19 V500 (HIB19 clone, BD Biosciences), anti-human CD137 APC (4B4-1 clone, Biolegend), anti-human CD69 Phycoerithryn(PE)-CF594 (FN50 clone, BD Biosciences), anti-human OX40 PE-Cy7 (Ber-ACT35 clone, Biolegend) and vitality dye Live/Dead Aqua (1:1000; eBioscience) to exclude dead cells. Activation was measured by the increase in CD137+OX40+ markers on CD4+ T cells and the increase in CD137+CD69+ markers on CD8+ T cells. All samples were acquired and analyzed on LSR Fortessa X-20 flow cytometry by FACS DIVA software 8.0 version (Becton Dickinson). In analyzing data from the AIM assays, the counts of AIM+ CD4+ and CD8+ T cells were normalized based on the counts of CD4+ and CD8+ T cells in each well to be equivalent to 1 × 10^6^ total CD8+ or CD4+ T cells. The Stimulation Index (SI) was calculated by dividing the percentage of AIM+ (CD4+CD137+OX40+/CD4+ and CD8+CD137+CD69+/CD8+) cells after SARS-CoV-2 stimulation with the ones in the negative control. 

For T6, we also tested the CD4+ and CD8 + T cell responses to alpha, beta, gamma, delta and epsilon variant Spike MPs.

### 2.4. Serological Analysis

To determine the level and durability of the humoral immune responses against SARS-CoV-2 infection, the quantitative antibodies (Abs) to Spike-1 protein (S) and to nucleocapsid protein (N) were detected with an electrochemiluminescence immunoassay. 

Anti-SARS-CoV-2 antibodies to the receptor binding domain (RBD) of the Spike-1 protein (anti-SARS-CoV-2 S) and to the nucleocapsid protein (anti-SARS-CoV-2 N) were detected with an electrochemiluminescence immunoassay (ECLIA; Elecsys^®^, Roche Diagnostics Ltd., Rotkreuz, Switzerland). Virus-specific total (IgG, IgA, IgM) antibodies are quantitatively measurable. The ECLIA uses a recombinant protein representing the RBD antigen in a double-antigen sandwich assay format, which favors the detection of high-affinity antibodies against the SARS-CoV-2 RBD protein. These antibodies have been shown to positively correlate with neutralizing antibodies in neutralization assays [24,25]. Anti-SARS-CoV-2 S antibodies are selectively elicited by the mRNA-based COVID-19 vaccines (Moderna and Pfizer-BioNTech), which have the RBD protein as the immunogen. Such antibodies are thus especially indicated to monitor the humoral immune response to vaccination with Moderna and Pfizer-BioNTech products, whereas anti-SARS-CoV-2 N antibodies are useful to identify subjects with potential concurrent-to-vaccination or past SARS-CoV-2 infection. The antibody levels, anti-RBD, anti-SARS-CoV-2 S (S Abs), were divided by 0.972 to report them in standard Binding Antibody Unit (BAU) [24]. The anti-SARS-CoV-2 N (N Abs) are reported in cutoff index (COI). 

### 2.5. Statistical Analysis

Characteristics of the included individuals were reported as a mean with standard deviation (SD) or number (N) and percentage (%), and to make comparisons between individuals with and without previous SARS-CoV-2 infection, we performed the Mann–Whitney test for age and the Chi-squared test or the Fisher’s exact test for the categorical variables. For the immunological values that were longitudinally measured, we performed linear mixed models with random intercept to evaluate changes over time. To compare individuals with and without previous SARS-CoV-2 infection, we used the Mann–Whitney test at every single timepoint and we used linear mixed models with random intercept to test the interaction between time and the presence of previous infection while adjusting for age and sex. Concerning the cellular immune responses that were available only after 6 months of the second dose, the Mann–Whitney test was used to compare the two groups. The level of significance was set to 0.05 and all the analyses were performed using Stata (v.16.0; Stata Corporation, College Station, TX, USA).

## 3. Results

The subjects were enrolled from December 2020 to October 2021. The blood samples were collected at different timepoints, as previously described. In group A (HCW without previous infection), 13/22 (59%) were female, compared to 5/7 (71%) in group B (previously infected). The mean age was 39.9 years (SD 13.1) versus 37.7 years (SD 13.2), respectively, as reported in Table 1. Only two subjects (one for each group) had comorbidities (hypertension, under treatment). No significant differences were observed between the two groups. The subjects in group B were vaccinated with the first dose of BNT162b2 vaccine at a mean of 8.6 months (SD 2.9) after SARS-CoV-2 infection. 

In group A, no subject developed a SARS-CoV-2 infection after first group allocation during the study period, and in group B no re-infections were documented. 

Concerning CD4+ T cell reactivity to Spike peptide megapools (MPS), the overall rate of change over time among the entire cohort of health workers resulted statistically significant at T1 (*p* = 0.038) and at T2 (*p* < 0.001) compared to T0, with a decrease at T6 (mean 7.96, SD 13.62, *p* > 0.05) (Figure 1).

As shown in Figure 2, the analysis of the CD4+ T cell response to Spike MPS was separately evaluated for group A and B. No significant differences were highlighted between the two groups at any timepoint. We only reported a trend of higher response in group A (subjects without previous infection). The overall time group interaction was not significant (*p* = 0.4016). 

CD8+ Spike reactivity showed a significant overall time group interaction (*p* = 0.0265) and also retained significance at each timepoint. Significant differences were also observed at the baseline (*p* = 0.0030) with higher SI in previously infected subjects, as expected (Figure 3). 

In the overall population, anti-S antibodies (S AbS) significantly increased at T1 vs. T0 (*p* = 0.0019), T2 vs. T0 (*p* < 0.001) and at T6 vs. T0 (*p* < 0.001) (data not shown).

Group B retained a significantly higher anti-S response at all the timepoints, including T6; the overall time group interaction was significant (*p* < 0.001), with a significant variation at all different timepoints (Figure 4, Panel A).

In Figure 4, Panel B, we measured AbS to nucleocapsid protein (N AbS) as the control, expecting to find it elevated only in previously infected individuals. Indeed, that was confirmed with our data suggesting that vaccinated individuals did not experience latent infections during the course of the study (*p* < 0.001). 

Finally, we assessed the capability of T cells to cross-recognize SARS-CoV-2 variants. As reported in Table 2, the AIM test was performed at T6 (six months after of the second dose) in both groups testing the CD4+ and CD8+ T cell responses against Spike MPs for five different SARS-CoV-2 virus variants (alpha, beta, gamma, delta, epsilon). No strong responses were detected for CD8+ T cells, while a high and retained CD4+ T cell response was observed to cross-recognize all tested variants including the epsilon variant, even if not detected as a circulant virus during the period of the study.

## 4. Discussion

The generation and clinical importance of T cell responses following SARS-COV-2 vaccination is discussed, particularly when focusing on protection against viral variants [4,26,27]. In our prospective cohort of HCW, we investigated the duration and effectiveness of cellular and humoral responses against SARS-CoV-2 vaccination in our overall cohort and in two distinct groups of subjects with and without previous infection. Information concerning the longevity of immunity to SARS-CoV-2 following natural infection may have considerable implications for the durability of immunity induced by vaccines. In fact, SARS-CoV-2-specific memory B and T cells persisted in the majority of patients up to 15 months after infections although a significant decrease in specific T cells, but not B cells, was observed between 6 and 15 months [28,29]. T cell responses are needed to support the high-affinity antibody production after vaccination and the induction of virus-specific CD4+ T cell responses, which exhibit a TH1 profile detectable by day 8 after priming, peak soon after vaccine boost and then fall to pre-boost levels after 4 months [6,30]. In our study, we found an increase in CD4+T cell response after the first and second dose of the BNT162b2 vaccine and a decrease after 6 months post full vaccination. The Spike-specific CD4+ T cell response did not show incremental differences between one or two vaccine doses, although a positive trend was observed in the group of subjects without previous infection. A more prominent decreased was additionally observed as a function of time in the infected followed by vaccination group. 

Spike-specific CD8+ reactivity was incremental as a function of the vaccination doses received and independent from previous exposure to the infection. We hypothesized that vaccination could help the CD8+ T cell reactivity specific for Spike peptides. As recently reported, a robust Spike-specific CD8+ T cell response is already elicited after the first vaccination. Fully functional vaccine-elicited early memory CD8+ T cells patrol the periphery for SARS-CoV-2 at least within the first months after vaccination [31,32]. The functional capacity of Spike-specific early memory CD8+ T cells is similar after vaccination and natural infection up to 3–4 months after boost or symptom onset [33]. Compared with natural infection, however, the early memory pool of Spike-specific CD8+ T cells after vaccination exhibits a different memory T cell subset distribution that may affect long-term maintenance characteristics [34,35]. In our cohort, the drop of CD8+ T cell activity in group B (previously infected subjects) could be dependent on several factors: (1) Study of T cell clones by genome-wide T cell epitope mapping reveals that expanded T cell clones with vaccination and infection occupied distinct space on single-cell maps, highlighting differences in the breadth of the epitopes recognized in vaccinated compared with infected individuals. (2) The most immunodominant epitopes recognized by CD8+ T cells in patients with COVID-19 are contained in ORF1 and not Spike protein. (3) Compared with natural infection, however, the early memory pool of Spike-specific CD8+ T cells after vaccination exhibits a different memory T cell subset distribution that may affect long-term maintenance characteristics. This difference may be caused by differential duration and location of antigen contact and different inflammatory responses after vaccination versus infection [5,36].

Conversely, our findings on Spike-specific CD8+ T cell follow-up in subjects without infection are in line with those of early efficacy studies and recent follow-up studies [5]. Antigen-specific CD8+ T cells have been reported to develop gradually and reach maximal levels only after the second dose and study of the magnitude of CD8+ T cell responses (measured using both AIM assay and intracellular cytokine responses) were both variable and several-fold lower in comparison to CD4+ T cell responses. This probably represents the maturation of the response in secondary lymphoid organs with subsequent release to the circulation.

The humoral response in our study revealed a persistent and higher antibody response in the overall cohort at any timepoint analyzed, with a significantly increasing trend especially among HCW who were previously infected with SARS-CoV-2. After a mean of about 9 months from natural infection, the nucleocapsid-specific AbS were still detected, with 7 times reduction at T6 when compared to T2 values and an 11 times reduction when compared to T0 values. 

In this group, the mean time elapsing from infection to vaccination was 8.6 months. This finding is indicative of persistent and prolonged humoral response in subjects previously infected within the prior 15 months. In contrast to CD8+T cells, peak mobilization of neutralizing antibodies and antigen-specific B cells to the periphery was first detectable after the second dose. This is in line with recent reports [37,38,39] and most probably represents the maturation of the response in secondary lymphoid organs with subsequent release to the circulation [40]. After the second dose, highly cross-neutralizing antibodies are present in the sera, clearly adding a major protective effector mechanism on top of the early mobilized Spike-specific CD8+T cell response. The humoral and CD8+T cell response are potentially coordinated by early elicited Spike-reactive CD4+T cells that underwent a limited boost expansion after the second dose of mRNA vaccination supporting their coordinating role [2]. Six months after the second dose in both groups, we tested the CD4+ T cell response to five virus variants to assess their cross-recognition capability. We found a persistent CD4+ T cell response able to cross-recognize circulating (alpha, beta, gamma, delta) and non-circulating (epsilon) variants, strengthening the cross-recognition capability over potential variant-specific exposure. This is in line with Tarke et al., who also found that the SARS-CoV-2-specific memory CD4+ and CD8+ T cells exposed to the ancestral strain by infection or vaccination effectively cross-recognize several variants of concern including omicron and showed that single amino acid substitution or deletions do not affect a polyclonal memory T cell response [41]. 

These data corroborated the message from our results that demonstrated a maintained CD4+ T cell response against non-circulating viral variants.

The most important limitation of our study is the small number of participants. 

We decided to enroll a particular subset of the general population, namely, healthcare professionals, which is often carried out in cohort studies (e.g., Nurses’ Health Study) because they are easier to follow-up on and they provide reliable information. However, healthcare professionals may be different in terms of rates of exposure to SARS-CoV-2 and general characteristics compared to the general population of working age and this may lead to problems of generalizability. Another limit of our study is the lack of comparison with subjects infected but not vaccinated and unvaccinated subjects who were not infected. Follow-up studies including larger cohorts of vaccines and SARS-CoV-2 convalescent vaccinated individuals are clearly required to assess the longevity of CD4+ and CD8+ T cell immunity. 

As reported in recent literature, it is possible that the sustained infection-acquired protection was affected by repeated low-dose occupational exposure to SARS-CoV-2 infection, enhancing protection against variants and inducing long-lasting memory T cell populations [42]. The SARS-CoV-2 vaccine efficacy remained high although it did decrease somewhat 6 months after full vaccination [13]. 

Analyses of data on cellular and humoral immune responses in this same cohort after nine months from the third vaccination dose and before the fourth dose are ongoing, to better define the durability of cellular and humoral response, in a risk population as well as HCW.

Since then, multiple variants of SARS-CoV-2 have been described, of which a few are considered variants of concern (VOCs), given their impact on public health. 

However, there is very limited information on the current situation of the circulating variants, such as genomics, transmissibility, treatment, management and efficacy of the vaccine [43].

These results related to the maintained immunological responses against non-circulating viral variants might help in the future to guide policies for the prioritization of a fourth vaccine dose in HCW. 

## 5. Conclusions

Altogether, our data indicate that humoral response was persistent and increased over time in HCW previously infected by SARS-CoV-2 when compared to uninfected individuals, up to 9 months of follow-up. The CD4+T cell response after vaccination was retained in both groups, while the CD8+ response was only maintained in previously infected individuals. At 6 months post-vaccination, CD4 + T cells were stable and able to cross-recognize circulating and non-circulating variants. Overall, SARS-CoV-2-specific responses after full vaccination are still able to recognize the virus and cross-recognize its variants but showed a progressive decline 6 months post-vaccination, which was more pronounced in uninfected individuals, suggesting the need for a booster dose.

## Figures and Tables

**Figure 1 vaccines-10-01784-f001:**
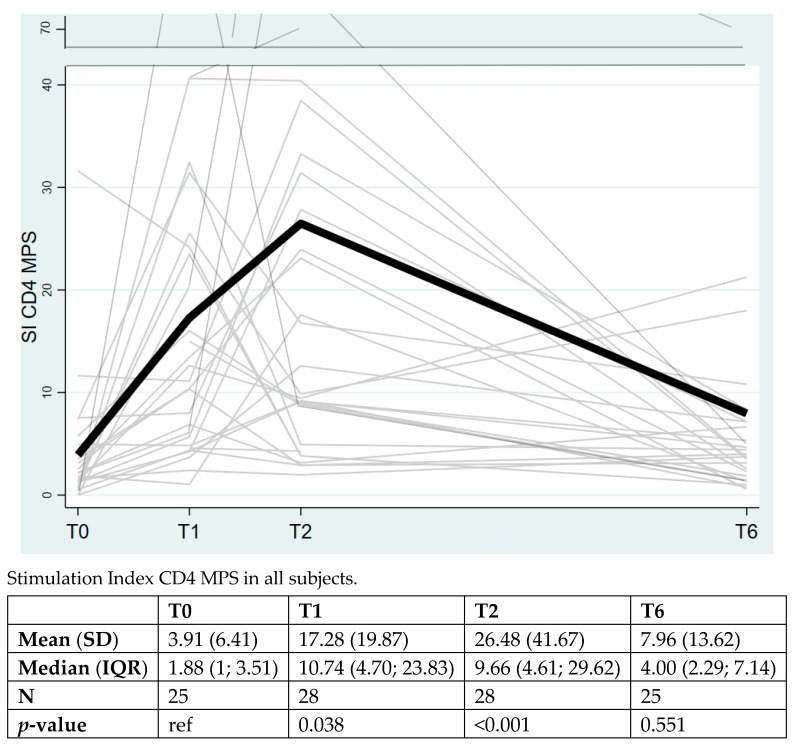
The overall rate of change over time for CD4+ T cell response to Spike megapools (MPS). The values were significant at T1 (*p* = 0.038) and at T2 (*p* < 0.001) compared to T0 with a decrease at T6 (*p* not significant). The data are expressed as Stimulation Index (SI), mean (standard deviation, SD) and median (interquartile range, IQR). The bolded line indicates the overall rate of change over time among the entire cohort of health workers.

**Figure 2 vaccines-10-01784-f002:**
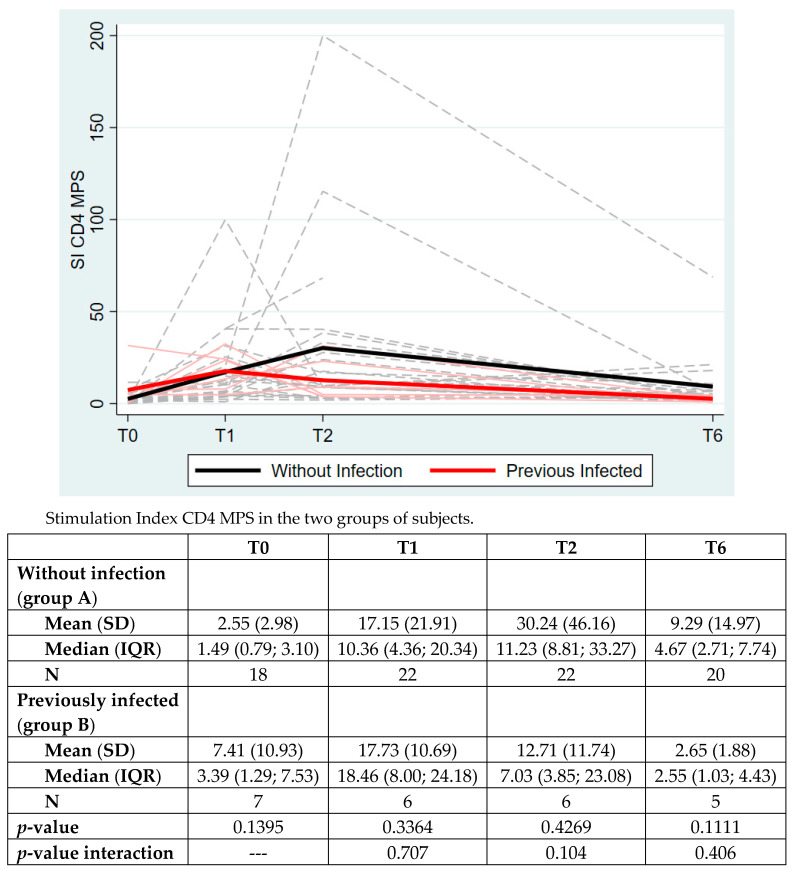
The CD4+ T cell response to Spike peptide megapools (MPS) in the two groups of subjects. The data are expressed as Stimulation Index (SI), mean (standard deviation, SD) and median (interquartile range, IQR). *p*-value between the two groups is analyzed at each timepoints with Mann–Whitney test.

**Figure 3 vaccines-10-01784-f003:**
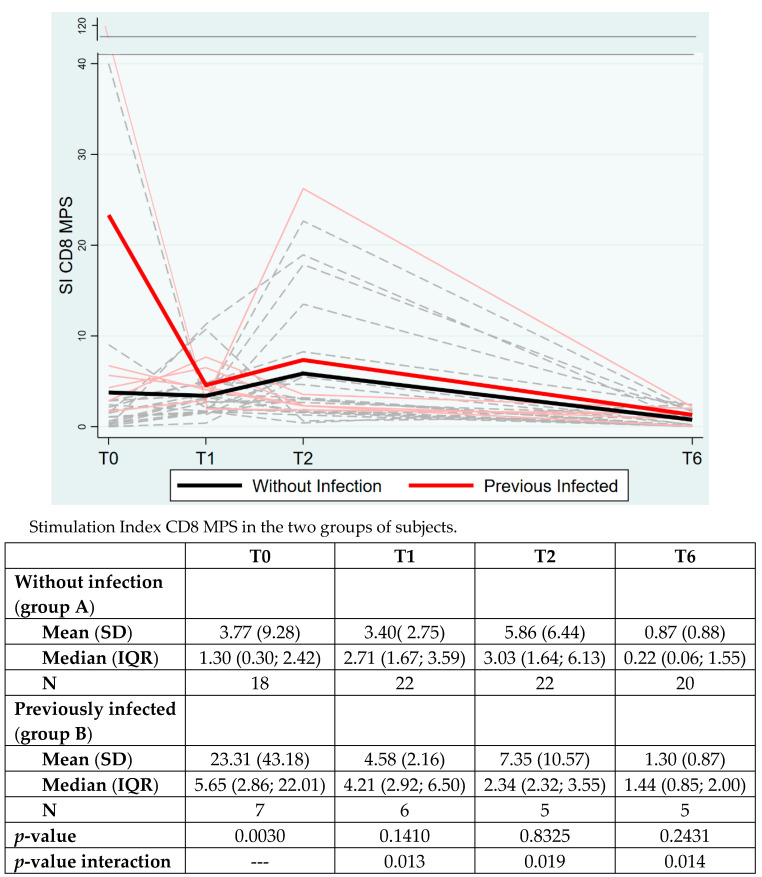
**The CD8+ T cell response to Spike peptide megapools** (**MPS**) **in the two groups of subjects**. The data are expressed as Stimulation Index (SI), mean (standard deviation, SD) and median (interquartile range, IQR). *p*-value between the two groups is analyzed with Mann–Whitney test at each timepoint.

**Figure 4 vaccines-10-01784-f004:**
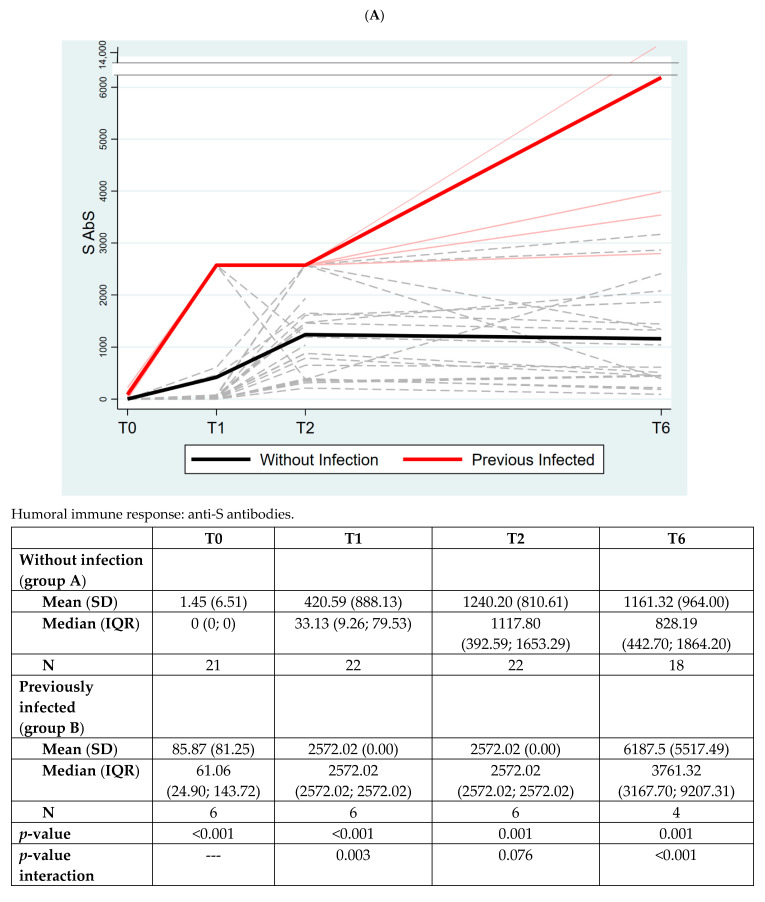
**Panel (A): The quantitative antibodies to Spike-1 protein (S AbS) in the two groups of subjects. Panel (B): The quantitative humoral immune response to nucleocapsid protein in the two groups of subjects.** The values of quantitative anti-S antibodies (S Abs) are expressed using Binding Antibody Unit (BAU) as mean (standard deviation, SD) and median (interquartile range, IQR). The values of quantitative anti-N antibodies (N Abs) are expressed using COI (cutoff index). *p*-value between the two groups is analyzed at each timepoint with Mann–Whitney test.

**Table 1 vaccines-10-01784-t001:** **Clinical characteristics of subjects enrolled in the study** (**overall population, group A and group B**)**.** The Mann–Whitney test was performed for age and the Chi-squared test or the Fisher’s exact test was performed for the categorical variables. SD: standard deviation. * Comorbidities: Hypertension, under therapy.

	Overall N = 29	Without Infection (Group A) N = 22 (76%)	Previously Infected (Group B) N = 7 (24%)	*p*-Value
**Female, N (%)**	18 (62.07)	13 (59.09)	5 (71.43)	0.677
**Age, mean (SD)**	39.38 (12.92)	39.91 (13.10)	37.71 (13.20)	0.970
**Comorbidities ***	2 (6.90)	1 (4.55)	1 (14.29)	0.431

**Table 2 vaccines-10-01784-t002:** Evaluation with activation-induced cell marker assay (AIM) of CD4+ and CD8+ T cell responses against different SARS-CoV-2 virus variants using Spike peptide megapools (MPS). The values of Stimulation Index (SI) are expressed as a mean (standard deviation, SD). The values are only available for T6 (six months after of the second dose). The Mann–Whitney test was performed. MPS: Spike peptide megapools.

	Overall N = 29	Without Infection (Group A) N = 22	Previously Infected (Group B) N = 7	*p*-Value
**CD4+ T cells**				
**T6_SI 4 MPS alpha**	4.91 (5.32) N = 24	5.09 (5.65) N = 20	4.00 (3.74) N = 4	0.862
**T6_SI 4 MPS beta**	6.11 (9.39) N = 25	7.00 (10.34) N = 20	2.55 (1.37) N = 5	0.371
**T6_SI 4 MPS gamma**	6.46 (9.30) N = 25	7.17 (10.26) N = 20	3.62 (2.42) N = 5	0.809
**T6_SI 4 MPS delta**	6.63 (11.84) N = 25	7.67 (13.06) N = 20	2.45 (1.80) N = 5	0.363
**T6_SI 4 MPS epsilon**	5.45 (6.25) N = 25	6.04 (6.85) N = 20	3.06 (1.77) N = 5	0.613
**CD8+ T cells**				
**T6_SI 8 MPS alpha**	2.03 (2.72) N = 24	2.16 (2.94) N = 20	1.39 (1.21) N = 4	0.477
**T6_SI 8 MPS beta**	1.56 (0.87) N = 25	1.66 (0.94) N = 20	1.18 (0.22) N = 5	0.530
**T6_SI 8 MPS gamma**	1.68 (0.82) N = 25	1.64 (0.80) N = 20	1.85 (0.98) N = 5	0.669
**T6_SI 8 MPS delta**	1.58(1.09) N = 25	1.51 (0.77) N = 20	1.86 (2.06) N = 5	0.669
**T6_SI 8 MPS epsilon**	1.76 (0.92) N = 25	1.80 (0.90) N = 20	1.58 (1.06) N = 5	0.621

## Data Availability

The raw data supporting the conclusions of this article will be made available by the authors, without undue reservation.

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
