# Peer review of "Healthcare Worker Study Cohort to Determine the Level and Durability of Cellular and Humoral Immune Responses after Two Doses of SARS-CoV-2 Vaccination"

_vaccines, 2022, doi:10.3390/vaccines10111784_

Round 1

Reviewer 1 Report

In this article, the authors studied immunological response against SARS-CoV-2 after vaccination among in health care workers without and with previous infection at different timepoints (T0, T1, T2 and T6). CD4+T cells response, CD8+T cells reactivity and specific antibodies were tested to evaluated the cellular and humoral immune-responses.

As the authors mentioned, the most important limitations of our study are the small number of participants and the lack of comparison with subject infected but not vaccinated and unvaccinated and uninfected. However, the article focused on health care workers, which may guide policies for prioritization of a 4th vaccine dose for them. Therefore, it still has its clinical application value.

We want to point out some revision advice so as to help readers, especially those who are not in the area of vaccines or immunity, understand the article better.

In Table 1, the clinical characteristics of each subtable needs to be mentioned or it will be confused to understand what does each date equal to.

In Figure 1, the upper part of the figure seems missing. It will be better to reupload the full picture to show all the data.

In Table 2, the value of MPS needs to be explained in the table or in the article to help readers how it can reflect the reactivity of CD4+T and CD8+T cells. Units of data can also be added in the table if necessary.

Author Response

In this article, the authors studied immunological response against SARS-CoV-2 after vaccination among in health care workers without and with previous infection at different timepoints (T0, T1, T2 and T6). CD4+T cells response, CD8+T cells reactivity and specific antibodies were tested to evaluated the cellular and humoral immune-responses.

As the authors mentioned, the most important limitations of our study are the small number of participants and the lack of comparison with subject infected but not vaccinated and unvaccinated and uninfected. However, the article focused on health care workers, which may guide policies for prioritization of a 4th vaccine dose for them. Therefore, it still has its clinical application value.

We want to point out some revision advice so as to help readers, especially those who are not in the area of vaccines or immunity, understand the article better.

In Table 1, the clinical characteristics of each subtable needs to be mentioned or it will be confused to understand what does each date equal to.

We thank the reviewer for this comment, which made us reconsider how to present results in order to make them more readable. For each figure, there is a table that shows descriptive information (mean (SD) and median (IQR)) and p-values. On the other hand, the figures show the mean values over time (same values reported in the tables) but also the spaghetti plot, thus showing the single patterns of each patient and consequently also the range of values at each time point. Even if the mean values are reported both in tables and in the plots, the information reported are thus not completely the same.  However, showing all the information reported in each table within the corresponding plot would make it difficult to read the results (too much information within the plot). As such, we decided to report both the tables and the figures but reporting each table and the corresponding plot within a single figure. We really think that this way of presenting our results can give the reader a clear and complete picture of the longitudinal pattern.

In Figure 1, the upper part of the figure seems missing. It will be better to reupload the full picture to show all the data.

We really thank the reviewer for this suggestion. We use this type of figure to include also the outlier data and to present the overall rate of change over time with a correct scale of data. We use the same presentation for figure 3 and 4.

In Table 2, the value of MPS needs to be explained in the table or in the article to help readers how it can reflect the reactivity of CD4+T and CD8+T cells. Units of data can also be added in the table if necessary.

Thank you for the comment. We reported all the data in the captions of the table, explaining the MPS and the measure of SI (as reported in the method section also)

‘Table 2. Evaluation with activation induced cell marker assay (AIM) of CD4+ and CD8+ T cells responses against different SARS-CoV-2 virus variants using Spike peptide megapools (MPS). The values of Stimulation Index (SI) are expressed as mean (standard deviation, SD). The values are available only for T6 (six months after of 2nd dose). The Mann-Whitney test was performed. MPS: Spike peptide megapools.’

Reviewer 2 Report

The humoral immune response which induced by vaccination or infection is the first line of body defense against SARS-CoV-2 infection. However, the rapid wanning of neutralizing antibody titer against SARS-CoV-2 over time may lead to reinfection. Therefore, the knowledge on the magnitude, timing and longevity of the cellular immunity against SARS-CoV-2 infection is vital for understanding the role that T cell response might play in disease clearance and protection from reinfection and/or disease. This study provides a valuable cohort evidence on cellular and humoral immune responses in health care workers (HCW) following vaccination. The MS could be accepted with minor modifications.

Minor comments:

1.     It is recommended that cellular and humoral immune outcomes, including CD4+ T cell responses, CD8+ T cell responses, humoral immune responses, and cross-activation against SARS-CoV-2 variants, be described separately in the Results section of this MS.

2.     The tables and figures for each cell marker or antibody should be combined because their content is the same.

3.     In the data, some mean of biomarker have larger SD compared to others such as in figure 3 and 4, consider using the median instead.

Author Response

The humoral immune response which induced by vaccination or infection is the first line of body defense against SARS-CoV-2 infection. However, the rapid waning of neutralizing antibody titer against SARS-CoV-2 over time may lead to reinfection. Therefore, the knowledge on the magnitude, timing and longevity of the cellular immunity against SARS-CoV-2 infection is vital for understanding the role that T cell response might play in disease clearance and protection from reinfection and/or disease. This study provides a valuable cohort evidence on cellular and humoral immune responses in health care workers (HCW) following vaccination. The MS could be accepted with minor modifications.

Minor comments:

  1. It is recommended that cellular and humoral immune outcomes, including CD4+ T cell responses, CD8+ T cell responses, humoral immune responses, and cross-activation against SARS-CoV-2 variants, be described separately in the Results section of this MS.

As you suggest, we maintained in the results section the separate description concerning the cellular (CD4+T and CD8+ T cells) and humoral (anti S and anti N) immune responses and at the end the cross-response against SARS-CoV-2 variants. Showing all the information reported in each table under figure within the corresponding plot would make it difficult to read the results (too much information within the plot). As such, we decided to report both the description tables and the figures but reporting each table and the corresponding plot within a single figure

  1. 2.  The tables and figures for each cell marker or antibody should be combined because their content is the same.

We thank the reviewer for this comment, which made us reconsider how to present results in order to make them more readable. For each figure, there is a table that shows descriptive information (mean (SD) and median (IQR)) and p-values. On the other hand, the figures show the mean values over time (same values reported in the tables) but also the spaghetti plot, thus showing the single patterns of each patient and consequently also the range of values at each time point. Even if the mean values are reported both in tables and in the plots, the information reported are thus not completely the same.  However, showing all the information reported in each table within the corresponding plot would make it difficult to read the results (too much information within the plot). As such, we decided to report both the tables and the figures but reporting each table and the corresponding plot within a single figure. We really think that this way of presenting our results can give the reader a clear and complete picture of the longitudinal pattern.

  1. 3. In the data, some mean of biomarker have larger SD compared to others such as in figure 3 and 4, consider using the median instead.

We thank the reviewer for underling the high variability in figure 3 and 4. For completeness, in accordance with the suggestion, we decided to report median (IQR) together with mean (SD) for all the variables reported.

Reviewer 3 Report

abstract:

the abstract could be more inclusive and organized. please consider refrasing the last sentence.

results:

table 1: for each segment of the table please specify what parameter is represented (CD4;CD8; antibody etc). - maybe add headings.

a sharp drop in CD8+ activity is noted for group B between T0-T1 , can you explain? is this finding discussed?

discussion:

lines 256-257 : regarding CD4+ activity level T1-T2 , what do you mean by increasing trend? 

discussion regarding findings on the CD8+ T cells - accordance or discordance with the cited references (ref.8)

conclusions:

conclusion not clear regarding CD8+ findings.

Author Response

abstract:

the abstract could be more inclusive and organized. please consider refrasing the last sentence.

We removed the last sentence ‘The vaccine could help CD8+ T cells reactivity specific for Spike peptides’ and we changed the abstract as follow:

Abstract: We prospectively studied immunological response against SARS-CoV-2 after vaccination among health care workers without (group A) and with previous infection (group B). The analyses were collected at T0 (before the BNT162b2), T1 (before the 2nd dose), T2 and T6 (1, 6 months after of 2nd dose). For cellular immune-response the activation induced cell marker assay was performed with CD4 and CD8 Spike peptide megapools expressed as Stimulation Index. For humoral immune-response we determine antibodies to Spike-1 and nucleocapside protein. The linear mixed model compared specific times to T0. The CD4+ Spike response overall rate of change was significant at T1 (p=0.038), at T2 (p< 0.001) decreasing at T6. For CD8+ Spike reactivity the interaction between time and group was significant (p=0.0265), and the P value for groups comparison was significant at baseline (p=0.0030) with higher SI in previously infected subjects.Overall, the anti-S Abs significantly increased from T1 to T6 compared to T0. The group B at T6 retained high anti-S titre (p< 0.001). At T6 in both groups we found a persistent humoral response and a high CD4+ T cells response able to cross recognize SARS-COV-2 variants including epsilon, even if not a circulating virus at that time.

results:

table 1: for each segment of the table please specify what parameter is represented (CD4; CD8; antibody etc). - maybe add headings.

We thank the reviewer for asking more clarifications regarding table 1. Now each segment of the table was reported together with the related plot in order to facilitate the reader's understanding of the tables. As such, we now have a figure for each parameter containing plot and table.

A sharp drop in CD8+ activity is noted for group B between T0-T1, can you explain? is this finding discussed?

We thank the reviewer for this important suggestion; we elaborate this concept and we add in the discussion this part.

The sharp drop of CD8+ T cell activity in Group B could be dependent from several factors:

  • Study of T cell clones by genome-wide T cell epitope mapping reveals that expanded T cell clones with vaccination and infection occupied distinct space on single-cell maps, highlighting differences in the breadth of the epitopes recognized in vaccinated compared with infected individuals
  • the most immunodominant epitopes recognized by CD8+ T cells in patients with COVID-19 are contained in ORF1 and not spike protein
  • Compared with natural infection, however, the early memory pool of spike-specific CD8+ T cells after vaccination exhibits a different memory T cell subset distribution that may affect long-term maintenance characteristics. This difference may be caused by differential duration and location of antigen contact and different inflammatory responses after vaccination versus infection, as indicated by a lower CD38 expression on early memory spike-specific CD8+ T cells after vaccination compared with natural infection (ref 8 and other new ref 36 Sureshchandra S, Lewis S A,  Doratt BM, Jankeel A, Coimbra Ibraim I, and Messaoudi I. Single-cell profiling of T and B cell repertoires following SARS-CoV-2 mRNA vaccineJCI Insight. 2021;6(24):e153201. https://doi.org/10.1172/jci. insight.153201)

discussion:

lines 256-257: regarding CD4+ activity level T1-T2, what do you mean by increasing trend? 

In lines 256-257 we reported that ‘The CD4+ T cells response to Spike MPS between the two groups did not show differences at timepoints, but an increasing trend in the group of subjects without previous infection was reported’. The increasing trend is related to the values reported in the table under figure 2 T1 mean 17.15, T2 30.24 and T6 9.29. The values were not significant, but the increasing mean (and median) was reported compared to values in group B T1 17.7, T2 12.7 and T6 2.6. This could indicate that the subjects with previous infection, also at baseline before vaccination, retain a memory response independently from second dose of vaccine. The responses decreased during time in this group of subjects with hybrid immunity in contrast to the individuals without prior infection, even if the time group interaction was not significant.

discussion regarding findings on the CD8+ T cells - accordance or discordance with the cited references (ref.8)

We added in the discussion section this part to better explain the findings on CD8+ T cells:

Concerning antigen-specific CD8+ T cell follow-up in Group A naïve subject, our findings are in line with those of early efficacy studies and recent follow-up studies, Antigen-specific CD8+ T cells have been reported to develop gradually and reach maximal levels only after the second dose  and study of  the magnitude of CD8+ T cell responses (measured using both AIM assay and intracellular cytokine responses) were both variable and several-fold lower in comparison to CD4+ T cell responses

This is in line with previous reports (ref 5, previous ref 8) and most probably represents maturation of the response in secondary lymphoid organs with subsequent release to the circulation.

conclusions:

Conclusion not clear regarding CD8+ findings.

We modified the conclusion as follow:

Taken together, our data indicate that humoral response was persistent and increased over time in HCW previously infected by SARS-CoV-2 when compared to uninfected individuals, up to 9 months of follow-up. The CD4+T cells response after vaccination was retained in uninfected subject. Considering the CD8+ Spike reactivity, we found higher values in previously infected subjects compared to those who were not infected, as a maintained memory response. We also detected for CD4 + T cells a capability to cross-recognize all tested variants especially epsilon variant, even if not detected as circulant virus during the period of the study.

Round 2

Reviewer 3 Report

1. some more english language editing is nedded

2. in  the section results:

lines: 213-215; 232-234; 265-265 - you can delete the explanation on statistical analysis as this had already been explained in material and methods section

Author Response

Reviewer 2:

  1. some more English language editing is needed

Thank you for your suggestion. One of our native English-speaking colleagues has corrected the manuscript. All the changes were highlighted in the text.

  1. in the section results:

lines: 213-215; 232-234; 265-265 - you can delete the explanation on statistical analysis as this had already been explained in material and methods section

 We agreed with your comments. We removed from the figure captions the sentences as indicated.